# Influence of Gas Pressure on the Failure Mechanism of Coal-like Burst-Prone Briquette and the Subsequent Geological Dynamic Disasters

Ying Chen [1,*], Zhiwen Wang [1], Qianjia Hui [1], Zhaoju Zhang [1], Zikai Zhang [1], Bingjie Huo [1], Yang Chen [2] and Jinliang Liu [3]

[1] Mining Engineering School, Liaoning Technical University, Fuxin 123000, China
[2] Research Center for Rock Burst Control, Shandong Energy Group Co., Ltd., Jinan 250014, China
[3] Huafeng Coal Mine, Xinwen Mining Group Co., Ltd., Taian 271413, China
* Correspondence: chenying@lntu.edu.cn

**Abstract:** Rock bursts and coal and gas outbursts are geodynamic disasters in underground coal mines. Laboratory testing of raw coal samples is the dominant research method for disaster prediction. However, the reliability of the experimental data is low due to the inconsistency of the mechanical properties of raw coal materials. The utilization of structural coal resources and the development of new coal-like materials are of significance for geodynamic disaster prediction and prevention. This paper studies the failure characteristics and dynamic disaster propensities of coal-like burst-prone briquettes under different gas pressures. A self-made multi-function rock–gas coupling experimental device was developed and burst-prone briquettes were synthesized, which greatly improved the efficiency and precision of the experimental data. The results showed that the burst proneness of the briquette was thoroughly reduced at a critical gas pressure of 0.4 MPa. When the gas pressure was close to 0.8 MPa, both the bearing capacity and the stored burst energy reduced significantly and the dynamic failure duration extended considerably, indicating the typical plastic-flow failure characteristics of coal and gas outbursts. The acoustic emission monitoring results showed that with the increase in gas pressure, the post-peak ringing and the AE energy ratio of coal samples increased, suggesting that the macroscopic damage pattern changed from bursting-ejecting of large pieces to stripping–shedding of small fragments adhered to mylonitic coal. In addition, the transformation and coexistence of coal failure modes were discussed from the perspectives of coal geology and gas migration. This study provides a new method for the scientific research of compound dynamic disaster prevention in burst coal mines with high gas contents.

**Keywords:** compound dynamic disasters; rock burst; coal and gas outburst; gas pressure; failure mode; briquette

## 1. Introduction

Geodynamic disasters in underground coal mines mainly include rock bursts and coal and gas outbursts. Both pose a serious threat to mining safety and present research areas in the underground coal mining industry [1,2]. The geoconditions that lead to their occurrence and appearance patterns are noticeably different, but they both solely occur in shallow underground mines. After years of research, prediction, monitoring, and prevention technology systems have been gradually established.

In recent years, with the increase in the mining depth, not only are geodynamic disasters more serious and frequent, but they have also become more complicated. For example, a large amount of abnormal gas emissions has been found to occur in coal bursts [3–5] and coal and gas outburst disasters have occurred in high burst-risk coal seams [6,7]. In addition, several coal seams have been identified as highly burst-prone, containing high

levels of gas [4–8]. These new geodynamic disaster phenomena are attributed to the deterioration of geological conditions, such as high ground stresses, high temperatures, and softening of the surrounding rock, and have brought great challenges to scientific research and to the current engineering practice of geodynamic disaster prediction and prevention in coal mines.

As two geodynamic disasters can undergo subsidiary transformations or even co-exist, a new research field has been established, namely compound dynamic disasters (CDDs), which include coal bumps along with abnormal gas emissions and rock bursts induced by coal and gas outbursts [8,9]. The study of CDDs focuses on the mutual influence, transformation, and interaction of the two kinds of dynamic disasters, as well as targeted prediction and prevention measures.

Two research methodologies are mainly employed to study compound dynamic hazards. One is case-based field studies, focusing on prediction and prevention technology corresponding to a site's specific geoconditions and mining arrangements and the reverse analysis of the mechanism of disaster occurrence [10,11]. For example, Zhao et al. [12] established a regional CDDs prediction and control technical system based on the characteristics of coal bursts and gas outbursts that occurred in the Pingdingshan coalfield. However, the research results relied highly on disaster data collection which cannot be carried out on a large scale. Another research method uses the theory of rock mechanics to investigate the occurrence process and key factors of disasters based on theoretical analyses or laboratory experiments [13–15]. These two research methods close the gap between theory and practice from the perspective of science and engineering.

For coal burst-dominated disasters, the change in the likelihood of coal bursts is mainly studied from the perspective of dynamic and static loading superposition or burst energy accumulation and dissipation. Dynamic and static load superposition is a traditional theory in rock burst research and has also been introduced into the CCDs research field in recent years [16–18]. One scientific work laid the foundation for prediction technology and determined the formula to calculate the burst tendency index of coal in compound dynamic disasters [19]. From the perspective of energy accumulation and dissipation, Song et al. [20] analyzed the energy of coal samples with a high impact tendency under different gas pressures under uniaxial and cyclic loads; however, the regularity of the results was weak due to data divergence. Xu et al. [21] proposed a modified method to determine coal burst proneness from the perspective of burst energy in cases of compound dynamic disasters dominated by coal bursts; however, only samples weakly prone to burst were involved. In short, the research to determine burst energy is still lacking in data.

Regarding the phenomenon of abnormal gas emissions after a coal burst, current research results show that this is mainly related to the micro-fracture structure of coal and rock, the amount of gas adsorption and analysis conditions, and the environmental temperature [3–5,21–24]. High ground stresses and high temperatures in deep coal seams induce compound dynamic disasters, and the effect of shock waves caused by a rock burst on the gas adsorption ability is the direct result of abnormal gas emissions.

In the research field of the fundamental theory of CDDs, some mechanical models have been established based on the gas–solid coupling constitutive method, and the critical state and influencing factors of the combination and mutual transformation of rock bursts and gas outbursts have been obtained. Noticeable research achievements include a mechanical model for CDDs in circular roadways [25]. Subsequent research on this topic [8] optimized the boundary conditions of the model. However, these theoretical analysis results can only be used as a reference, as they make various assumptions, and it is difficult to measure the relevant parameters.

In recent years, experimental research on CDDs has mainly focused on the impact of gas on burst propensity. However, due to the use of raw coal samples, conclusions drawn from these experimental results are quite different [20,26,27]. Acoustic emission (AE) technology has been widely used in the study of single disasters of rock bursts and

coal and gas outbursts [28,29]; however, the effect of gas pressure on the microstructure damage and fracture development of the coal body has not yet been determined.

In this study, the transformation and combination of dynamic disasters under different gas pressures are analyzed using experimental methods. A multi-function solid–gas coupling device was developed and burst-prone briquette specimens were used to improve the reliability and repeatability of the experimental results. The effect and influence of gas pressure on the burst characteristics of coal samples are explored. The energy evolution, catastrophic tendency, and mechanical mechanism of coal deformation under different gas pressures are further explored based on the AE characteristics of coal samples under loading conditions. This study provides new experimental methods and analytical techniques for the study of CDDs and provides the basic theory for the engineering practice of disaster reduction and prevention in deep coal mines with high gas contents.

## 2. Materials and Methods

### 2.1. Mold Design

In situ coal samples normally have internal micro defects and joints, leading to a huge inconsistency between samples. In addition, coal with a low burst proneness tends to be soft and heavily fractured, resulting in significant difficulties regarding sample testing. As such, in order to reduce the inconsistency between the testing samples, as well as the scatter in testing data, homogeneous coal briquettes with controlled strengths were prepared for testing.

A 50 mm × 100 mm standard size cylindrical mold was built for producing specimens. The mold consists of a compressive arm, two semi-circular steel tubes, a bottom plate, and three fixing devices to bolt the two semi-circular steel tubes at three different positions (Figure 1). The bottom plate is placed at the bottom of the cylindrical tube. Once the coal powder is poured into the cylindrical mold, the compressive arm is pulled down to compact the powder until the coal specimen is successfully formed. All the components of the mold are made of stainless steel. A schematic diagram of the mold is shown in Figure 1.

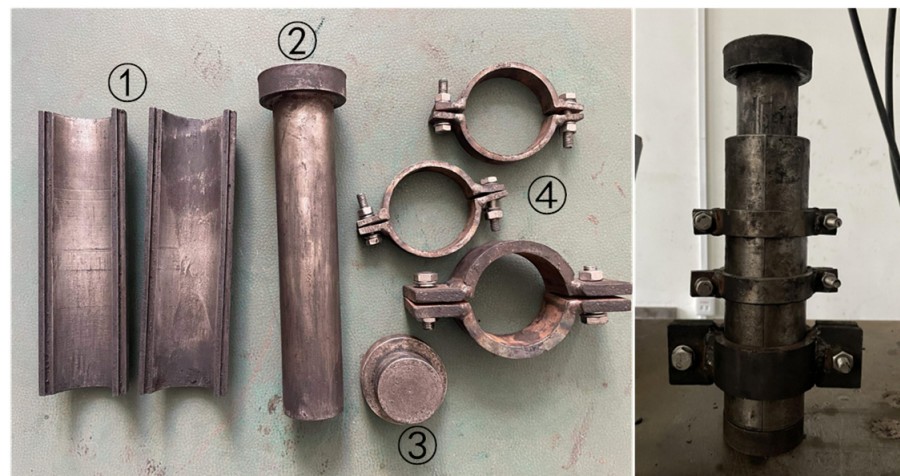

**Figure 1.** Specimen-producing mold. ① Semi-circular steel tube, ② compressive arm, ③ bottom plate, ④ fixing device.

### 2.2. Specimen Preparation

In order to prepare a coal specimen with appropriate strength and burst proneness, the coal powder and a binder were mixed under appropriate conditions. The coal powder was crushed and ground such that the particle diameter was below 0.3 mm. Then, it was mixed with PS32.5 cement, Hypromellose, and sodium humate powder in different ratios.

As a result of extensive laboratory tests to determine the optimal mixing ratio, the coal specimen was prepared with coal powder, cement, hypromellose, sodium humate

powder, and water in the following ratio: 0.9:1.7:0.3:0.3:0.4. The compressive strength of a coal specimen with such a ratio is 7–8 MPa.

The specimen preparation process is shown in Figure 2. The specific process was as follows:

Step 1: The coal powder was ground until all particles had a diameter of less than 0.3 mm and then the coal powder was mixed with cement, hypromellose, sodium humate powder, and water at the optimal ratio mentioned above.

Step 2: The mixture was poured into the mold up to a height of 2 cm and then the material was compressed under 50 kN of compressive stress for 10 min. This process was repeated until the height of the specimen was 10 cm.

Step 3: Once the specimen was successfully formed, it was compressed using the compressive arm and then the sample was removed from the mold.

Step 4: The specimen was placed in a 50 °C oven for over 48 h to dry and harden.

The mixing ratio, mold, pressing stress (50 kN), drying, and hardening environment used in the sample preparation are all the same, so the physical and mechanical properties of the specimen are highly consistent.

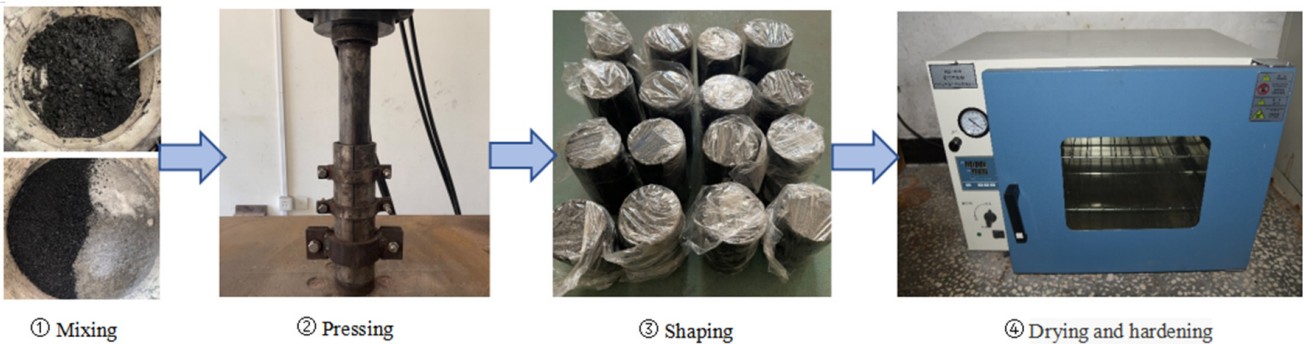

**Figure 2.** Procedure for coal specimen preparation.

### 2.3. New Multi-Phase (Solid–Gas Coupling) Testing Facility

The new testing facility consisted of the following components. (1) A servo control loading system with a solid–gas coupling device. The maximum vertical loading force of the servo control loading system is 600 kN. The solid–gas coupling device is a sealed chamber that has a maximum gas pressure of 20 MPa. The function of this component is to apply vertical stress to the specimen under a given gas pressure. (2) A gas supply system with a vacuum pump and a gas cylinder. The gas cylinder can provide a maximum gas pressure of 15 MPa. The maximum pressure of the vacuum pump is less than 0.06 Pa. The function of this component is to place the solid–gas coupling device under vacuum and inject gas into the chamber to achieve adsorption saturation of the specimen under a given gas pressure. (3) An acoustic emission (AE) monitoring system. During the experiment, an acoustic emission signal sensor was installed on the surface of the specimen. The acoustic emission signal generated during the loading process was collected and stored through the AE monitoring system. (4) An external monitoring system. This component uses high-speed cameras to record the process of specimen failure. The multi-phase (solid–gas coupling) testing facility has been granted a patent by China National Intellectual Property Administration. The laboratory testing system is illustrated in Figure 3.

### 2.4. Testing Scheme

The solid–gas coupling testing chamber containing the coal specimen was placed under vacuum before the test in order to inject pure gas. The gas pressure was monitored during gas injection until the pressure reached the required level. In this study, identical experiments were conducted under gas pressures of 0.2, 0.4, 0.6, 0.8, and 1 MPa in order to conduct a comparative analysis. Then, the chamber was left for 24 h to allow the specimen to be fully saturated with gas. Once the specimen was ready, it was tested with the compression machine. Note that the gas pressure was maintained at a constant level

for the duration of the test. Burst proneness indicators included the uniaxial compressive strength, the duration of dynamic fracture, and the bursting energy index. In the meantime, an AE monitor was employed to capture the continuous change in AE counts and energy. The detailed procedure for testing was as follows:

Step 1: The coal specimen was placed into the solid–gas coupling testing chamber. An AE sensor was attached to the coal specimen surface within the solid–gas coupling testing chamber. Gas was then injected into the chamber until the gas pressure reached 0.2 MPa.

Step 2: After the coal specimen was immersed in the gas at the required gas pressure for 24 h, burst proneness tests were conducted according to the GB/T25217.2-2010 [30] standard to measure the uniaxial compressive strength, the duration of dynamic fracture and the bursting energy index.

Step 3: The previous two steps were repeated to measure the burst proneness indicators under different gas pressures (0.4, 0.6, 0.8, and 1 MPa).

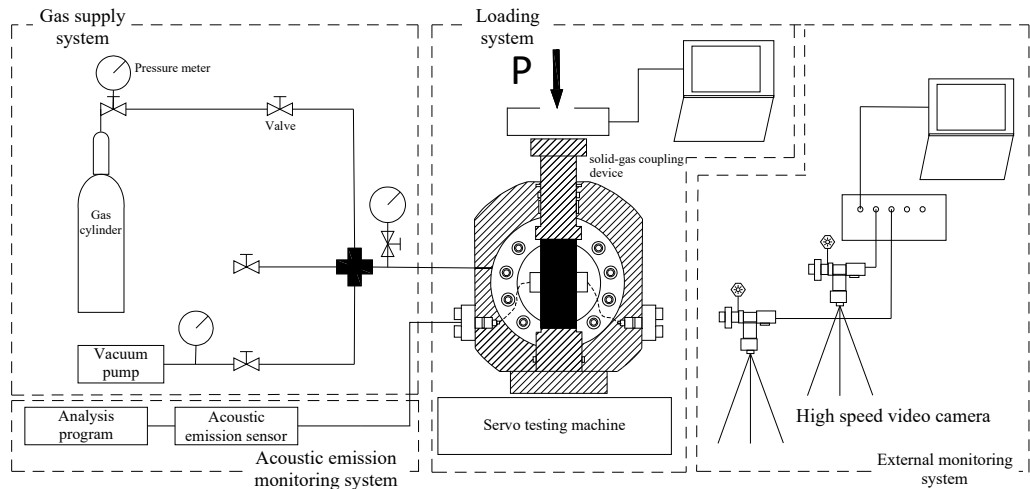

**Figure 3.** Schematic diagram of the new multi-phase testing facility.

## 3. Results

### 3.1. The Effect of Gas on Coal Failure Modes

The macroscopic damage patterns of coal specimens at various gas pressures ranging from 0 to 1 MPa were observed (Figure 4). With no gas present, single-slope shear failure occurred in the specimen, and it was broken into larger blocks that were ejected at the moment of failure. In contrast, when gas was present, more cracks were generated as a result of loading and the size of the cracked bulk tended to decrease. When the gas pressure was 0.2 MPa, some mylonitized coal was observed. When the gas pressure was increased up to 0.4 MPa, the number of cracks increased while the size of the cracks decreased. In the meantime, a large amount of mylonitized coal was formed. The failure of the coal specimen was mainly characterized by the stripping and shedding of small fragments. The test results show that with the increase in gas pressure, the macroscopic damage pattern of coal gradually changes from bursting-ejecting of large pieces to stripping–shedding of small fragments attached to mylonitic coal.

According to the test results, it is hypothesized that the influence of gas on the mechanical properties, failure mode, and even catastrophic characteristics of coal is the combined result of the mechanical response of the inherent properties of coal to an external load and the adsorption softening of gas on the surface of coal matrix particles. There is a critical gas pressure for this effect. When the gas pressure is lower than the critical value, the macroscopic mechanical response of coal is affected by its inherent properties. When the gas pressure exceeds the critical value, the mechanical properties of coal are gradually

affected by the gas. Thus, it is very important to determine the critical gas pressure to be able to control the change in the mechanical properties of coal.

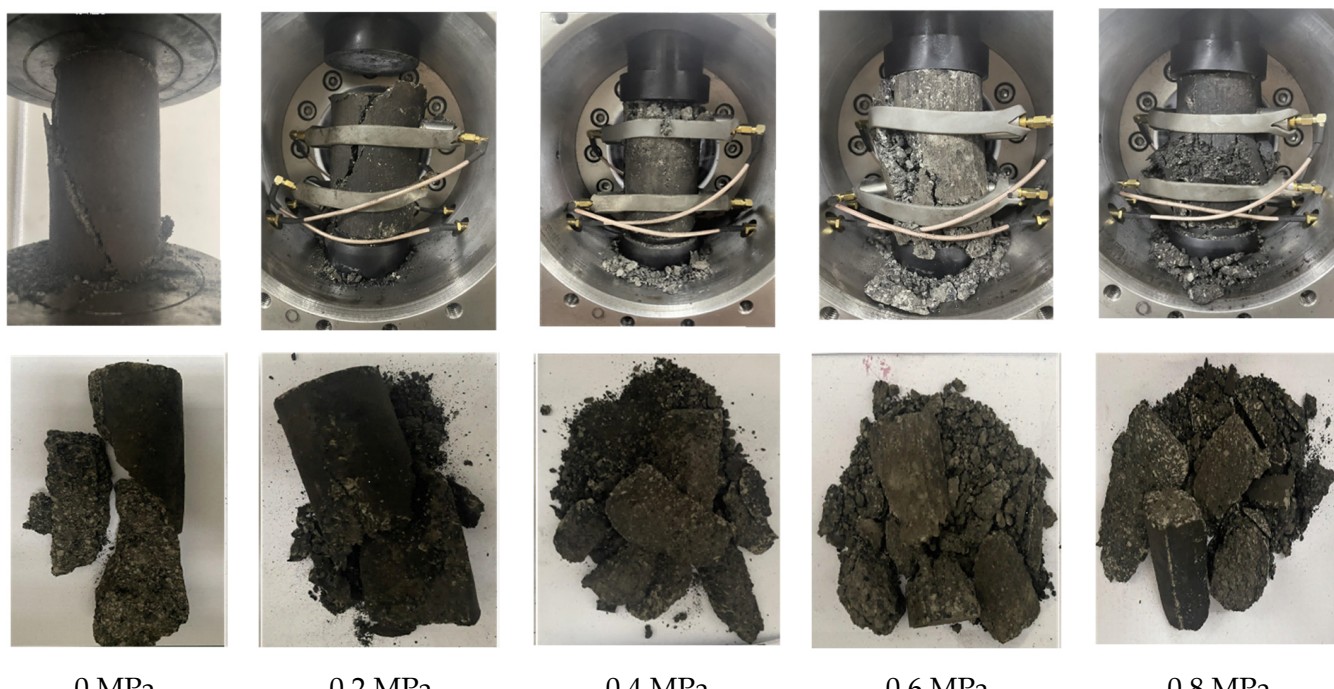

| 0 MPa | 0.2 MPa | 0.4 MPa | 0.6 MPa | 0.8 MPa |

**Figure 4.** Macroscopic damage pattern of coal specimens under different gas pressures.

*3.2. The Effect of Gas on Burst Proneness*

According to the national standards of the People's Republic of China (GB/T 25217.2-2010) [30], "methods for testing, monitoring and prevention of rock bursts, part 2: classification and laboratory test method on bursting liability of coal", the burst proneness of a coal specimen can be comprehensively measured according to four indexes, including the uniaxial compressive strength, the duration of dynamic fracture, the elastic strain energy index and the bursting energy index. The burst proneness of coal is classified according to its index value (Table 1).

**Table 1.** Criterion of burst proneness of coal.

| | Category | I | II | III |
|---|---|---|---|---|
| | **Burst Proneness** | **None** | **Weak** | **Strong** |
| Index | Duration of dynamic fracture $DT$/ms | $DT > 500$ | $50 < DT \leq 500$ | $DT \leq 50$ |
| | Elastic strain energy index $W_{ET}$ | $W_{ET} < 2$ | $2 \leq W_{ET} < 5$ | $W_{ET} \geq 5$ |
| | Bursting energy index $K_E$ | $K_E < 1.5$ | $1.5 \leq K_E < 5$ | $K_E \geq 5$ |
| | Uniaxial compressive strength $R_c$/MPa | $R_c < 7$ | $7 \leq R_c < 14$ | $R_c \geq 14$ |

### 3.2.1. Uniaxial Compressive Strength

The uniaxial compressive strength refers to the ratio of the failure load of the coal specimen under uniaxial compression to the area of its loading surface. The compressive strength variation curve of coal specimens under different gas pressures is illustrated in Figure 5. The maximum compressive strength of coal specimens without any gas present was 8.38 MPa, which is higher than that with the gas present. This indicated that the burst proneness of coal specimens without gas is weak according to Table 1. At a gas pressure of 0.2 MPa, the compressive strength was reduced by 8.5% to 7.66 MPa. The coal specimen is still weakly burst-prone under these conditions. On the other hand, when the gas pressure was increased to 0.4 MPa, the coal compressive strength was reduced by 28.8% to 5.96 MPa.

The compressive strength was further reduced by 36% to 5.36 MPa when the gas pressure was increased to 0.6 MPa. At gas pressures of 0.8 and 1 MPa, the compressive strengths were 4.99 and 4.60 MPa, exhibiting a 40% and 45% reduction, respectively. As a result, it is apparent that gas absorption could reduce the compressive strength of coal. Considering only the compressive strength index (based on Table 1), the burst proneness of the coal specimen is lost at a critical gas pressure of 0.4 MPa.

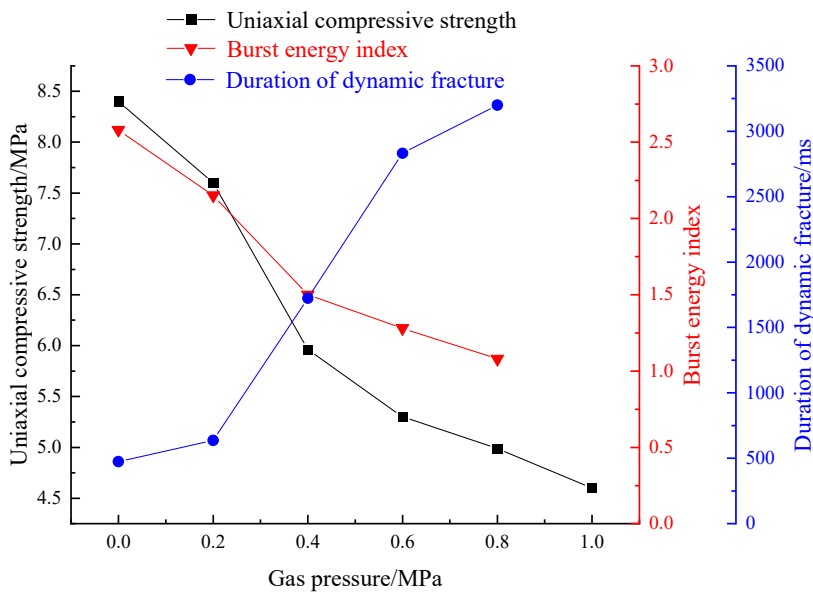

**Figure 5.** Change in burst proneness at different coal seam gas pressures.

Under gas pressure, expansion stress is generated inside the sample. This weakens the bonding among matrix particles and promotes the development of internal cracks in the sample, resulting in higher damage values at peak strain [31]. This reduces the coal's bearing capacity and increases its softness.

### 3.2.2. Duration of Dynamic Fracture

The duration of dynamic fracture refers to the time from ultimate strength to complete failure of a coal specimen under uniaxial compression. Figure 5 shows the duration of the dynamic fracture variation curve under different gas pressures. It can be observed that the duration of dynamic fracture increased from 475 ms to 3540 ms as the gas pressure increased from 0 MPa to 1 MPa. When the gas pressure was 0.2 MPa, the duration of the dynamic fracture of the specimen was 673 ms, indicating a loss of burst proneness (based on Table 1). It is therefore concluded that the duration of dynamic fracture increases with the increase in gas pressure. This leads to the gas-saturated coal specimen changing from brittle to tough. The increase in the duration of the dynamic fracture indicates that after the stress peak, the failure process and energy release become slow and moderate, which inhibits bursting. Considering only the duration of dynamic fracture (based on Table 1), the burst proneness of the coal specimen is lost at a critical gas pressure of 0.2 MPa.

### 3.2.3. Burst Energy Index

The burst energy index refers to the ratio of the accumulated energy before the stress peak to the consumed energy after the stress peak under uniaxial compression. The burst energy index variation curves under different gas pressures are illustrated in Figure 5. It can be observed that the burst proneness tends to decrease as the gas pressure increases, and when there is no gas present, the burst energy index is 2.58, indicating a weak burst proneness (according to Table 1). Following a continuous increase in gas pressure to 0.2 MPa and 0.4 MPa, the burst energy indices were 2.26 and 1.52, respectively, indicating that the coal is burst-prone in both scenarios. Finally, the burst energy indices were 1.28

and 1.08 at gas pressures of 0.6 and 0.8 MPa, respectively, which indicates that the coal is not burst-prone in both of these scenarios. Considering only the burst energy index (based on Table 1), the burst proneness of the coal specimen is lost at a critical gas pressure of 0.6 MPa.

The calculated curve of the burst energy index under different gas pressures is shown in Figure 6 (taking no gas and a gas pressure of 0.6 MPa as an example). Without gas, the accumulated strain energy before the stress peak is equal to the area surrounded by $0C_0Q_0$, and the consumed energy after the stress peak is equal to the area surrounded by $Q_0C_0D_0F_0$. Under a gas pressure of 0.6 MPa, the accumulated strain energy before the stress peak is equal to the area surrounded by $0C_{0.6}Q_{0.6}$, and the consumed energy after the stress peak is equal to the area surrounded by $Q_{0.6}C_{0.6}D_{0.6}F_{0.6}$. It can be observed from Figure 7 that under no gas, the stress increases rapidly after loading. The slope of the curve clearly indicates the hard characteristics of the specimen. The accumulated strain energy is more than that of gas-saturated coal (the area surrounded by $0C_0Q_0$ is larger than that surrounded by $0C_{0.6}Q_{0.6}$). After the stress peak, the stress–strain curve suddenly and sharply decreases, which indicated that the specimen quickly loses bearing capacity, resulting in a strong brittle bursting failure. At the same time, it also shows that the failure after the stress peak consumes less energy, and a large amount of energy is transferred to the kinetic energy of the fragments. Larger fragments are generated after the specimen is destroyed, and ejection occurs at the moment of destruction. With the increase in gas pressure, the strain of the coal specimen increases quickly, and the load capacity decreases significantly after loading, which demonstrates obvious softening compared to the coal specimen that does not contain gas. The accumulated strain energy is less than that of the coal specimen that does not contain gas. After the stress peak, the stress–strain curve decreases gently, indicating a plastic flow failure, which further indicates that the energy consumed after the stress peak increases (the area surrounded by $Q_{0.6}C_{0.6}D_{0.6}F_{0.6}$ is larger than that surrounded by $Q_0C_0D_0F_0$) and the surplus energy used to eject the fragments decreases. As the gas pressure increases, the fragmentation of the specimen increases. The damage pattern of the specimen changes from ejection to stripping–falling.

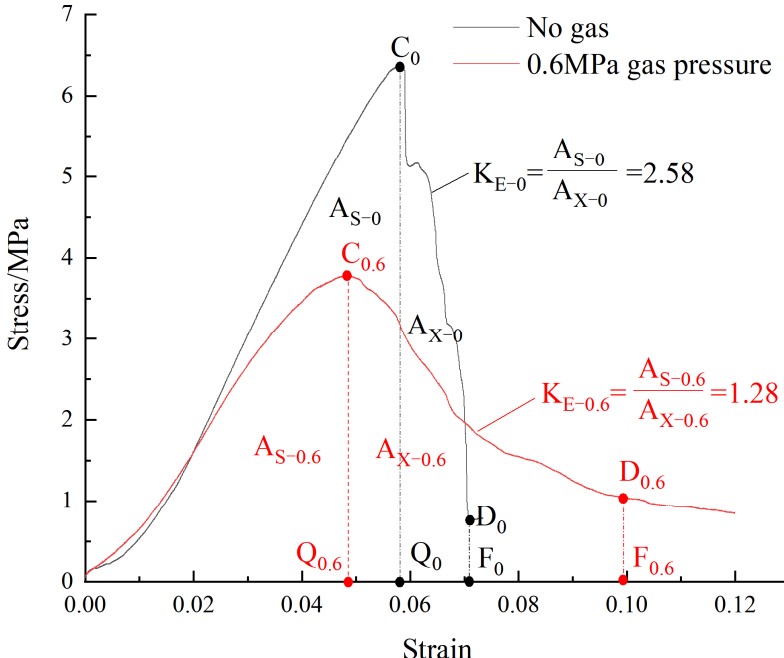

**Figure 6.** Calculated curve of the burst energy index under different gas pressures.

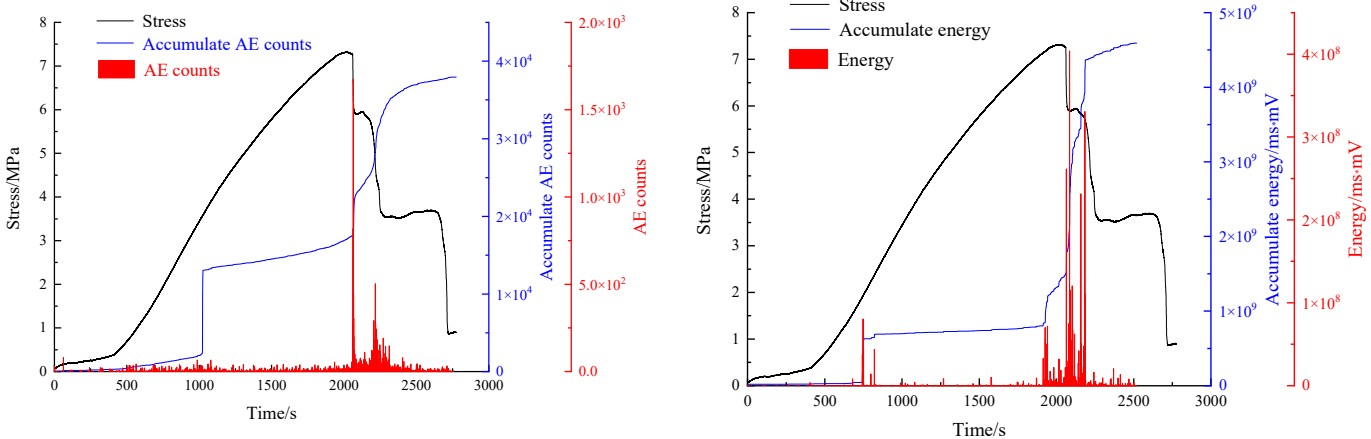

**(a)** AE counts and accumulated counts

**(b)** AE energy and accumulated energy

**Figure 7.** AE counts and energy characteristics under no gas.

### 3.3. AE Data Analysis for Gas-Saturated Coal Specimen

The AE counts and energy characteristics of the specimen during the loading process are demonstrated in Figures 7 and 8. Significant AE counts were recorded when the axial loading was close to the stress peak, which indicates that the formation of internal micro-cracks was initiated at the end of the elastic deformation stage. As the stress continued to increase, the cracks propagated until they were fully developed throughout the whole specimen. The maximum AE count and energy appear before and after the stress peak, which illustrates that the failure and released energy are most significant near the stress peak.

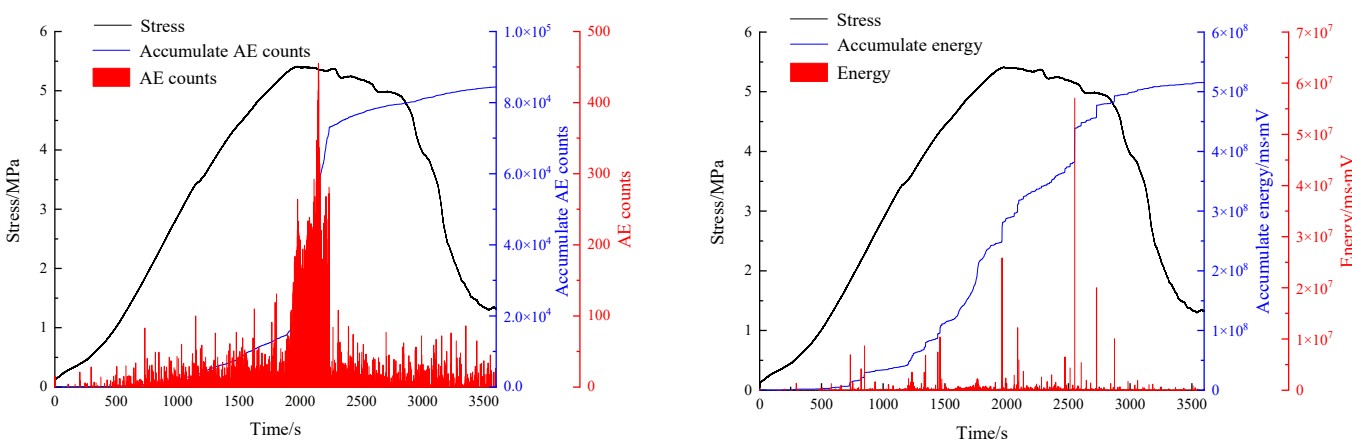

**(a)** AE counts and accumulated counts

**(b)** AE energy and accumulated energy

**Figure 8.** AE counts and energy characteristics under a 1 MPa gas pressure.

The distributions of AE counts and energy under different gas pressures were statistically analyzed, and the results are shown in Figures 9 and 10. When the gas pressure increased from 0 to 1 MPa, the maximum AE count dropped from 1673 to 455 and the corresponding maximum energy dropped from $4.04 \times 10^8$ ms·mV to $5.71 \times 10^7$ ms·mV. It is also noteworthy that under the influence of the gas, an increased AE count and energy were observed during the residual strength stage of the coal specimen compared to those without gas. The proportion of AE counts after the stress peak increased from 4.4% to 24.2%. The proportion of AE energy after the stress peak increased from 1.3% to 27.6%.

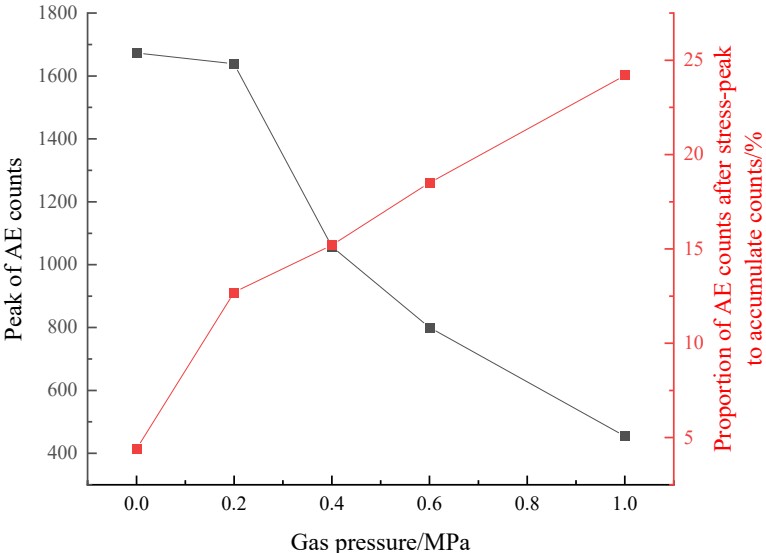

**Figure 9.** AE count characteristics under different gas pressures.

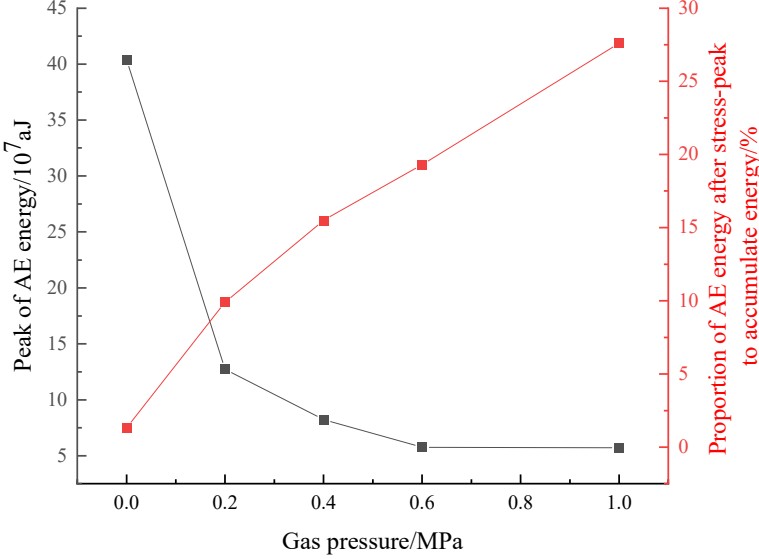

**Figure 10.** AE energy characteristics under different gas pressures.

Under gas pressure, the amount of gas attached to the surface of the coal matrix particles increases and the cementation continuity between the matrix particles weakens. The irregular micro-strain of the matrix produced by the adsorption of gas leads to relative dislocation between particles [22] and causes the development of micro-cracks, which increases the initial damage to the bearing structure. After a load is placed on the specimen, the damage is further aggravated, which inhibits the accumulation of strain energy. This is the reason why the AE counts and energy are lower at the stress peak of the specimen. With the increase in gas pressure, the proportion of AE counts and energy increases after the stress peak, which indicates that sustained plastic failure occurs in the specimen. This consumes most of the energy, and thus the energy used to produce bursting–ejecting failure decreases. This is the reason why the burst energy index of the specimen decreases with the increase in gas pressure.

## 4. Discussion

Burst proneness is a natural property of coal. The uniaxial compressive strength, the duration of dynamic fracture, and the burst energy index are important indices to

characterize this property. Accurately obtaining their values is of great significance to the prediction and prevention of rock bursts. With the gradual deepening of coal mining, the high-pressure gas contained in coal seams becomes a significant factor that cannot be ignored in rock burst prevention. In this paper, the variation in macroscopic failure characteristics, the compressive strength, the duration of dynamic fracture, and the burst energy index of coal with the increase in gas pressure was determined by extensive testing. The results showed that gas alters the inherent burst proneness of coal and also affects the occurrence potential of rock bursts.

The gas pressure and gas content in deep coal seams are generally high. High gas pressures have a significant impact on the burst proneness of coal, by weakening the strength of coal, prolonging the dynamic failure duration, reducing the burst energy index, and leading to increased strain energy consumption after the stress peak of the specimen. The failure mode of coal changes from brittle bursting to plastic flow. The absorbed gas softens the coal, leading to smaller broken pieces and the appearance of mylonitic coal. Gas desorption causes a large amount of free gas to accumulate, which provides prime conditions for the rock burst to induce a gas outburst. Therefore, deep, high-gas-content coal seams have a lower critical index of catastrophe and an indistinct propensity to cause disasters. These coal seams are easily disturbed by mining, which can induce rock bursts, gas emissions, and outbursts. Destructive damage to the production environment is the result of compound disasters.

This paper analyzed the influence of gas pressure on the failure mechanisms and the burst proneness of coal-like briquettes which are only weakly prone to bursting. The relationship between gas pressure and the mechanical properties of coal samples with a high burst proneness needs to be further studied.

## 5. Conclusions

(1) A new multi-phase (solid–gas coupling) testing facility composed of a servo control loading system with a solid–gas coupling device, a gas supply system, an acoustic emission (AE) monitoring system, and an external monitoring system has been developed. The preparation of burst-prone coal briquettes was performed for the first time. This provides an experimental basis to determine the influence of gas on the failure mode and burst proneness of specimens.

(2) The gas content has a great influence on the failure mode of coal specimens. With an increase in gas pressure, the macroscopic damage pattern of coal specimens gradually changes from bursting-ejecting of large pieces to stripping–shedding of small fragments adhered to mylonitic coal.

(3) The gas pressure affects the burst proneness index of coal specimens and causes the weakening and loss of the burst properties of coal specimens. The compressive strength of the coal specimen decreases with the increase in gas pressure, and the critical gas pressure for the loss of burst proneness is 0.4 MPa. The duration of dynamic fracture of coal specimens increases with the increase in gas pressure, and the critical gas pressure for the loss of burst proneness is 0.2 MPa. The burst energy index decreases with the increase in gas pressure, and the critical gas pressure for the loss of burst proneness is 0.6 MPa. The duration of dynamic fracture is more sensitive to gas pressure.

(4) The ratio of the AE counts and energy after the stress peak to the accumulated AE counts and energy increases with the increase in gas pressure, which shows that the consumed energy after the stress peak increases. This eventually leads to the loss of burst proneness of the coal specimen.

(5) Based on the test results, it is suggested that when identifying the burst proneness of deep, high-gas-content coal seams, it is necessary to account for the corresponding gas pressure in the environment in the relevant burst proneness tests.

**Author Contributions:** Methodology, Y.C. (Ying Chen); Validation, Z.Z. (Zhaoju Zhang); Formal analysis, Z.W. and Q.H.; Investigation, B.H.; Resources, J.L.; Data curation, Z.W.; Writing—original draft, Y.C. (Ying Chen); Writing—review and editing, Y.C. (Ying Chen); Visualization, Z.Z. (Zikai Zhang); Funding acquisition, Y.C. (Yang Chen). All authors have read and agreed to the published version of the manuscript.

**Funding:** This research was funded by the National Natural Science Foundation of China (Grant No. 51874164), Major Scientific and Technological Innovation Projects in Shandong Province (Grant No. 2019SDZY02), and the Science and Technology Plan Announced Bidding Project in Shanxi Province (Grant No. 20191101015).

**Institutional Review Board Statement:** Not Applicable.

**Informed Consent Statement:** Not Applicable.

**Data Availability Statement:** The data used to support the findings of this study are available from the corresponding author upon reasonable request (chenying@lntu.edu.cn).

**Conflicts of Interest:** The authors declare no conflict of interest.

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
