# Peer review of "Influence of Gas Pressure on the Failure Mechanism of Coal-like Burst-Prone Briquette and the Subsequent Geological Dynamic Disasters"

_sustainability, doi:10.3390/su15107856_

Round 1
Reviewer 1 Report
The manuscript describes relevant results of testing several controlled briquette specimens in order to identify the influence of gas pressure on failure mechanism of these specimens.
It is a good manuscript; therefore I have only the following suggestions:
Include a short explanation about why you determined that the ratio in line 136 was the optimal mixing ratio.
Please verify the word “unform” in line 117.
Please add the letter “s” to the word Figure in line 288.
Author Response
Attachment

Reviewer 2 Report
With the increase of mining depth, the dynamic disasters become more serious, and the topic is significant. This paper developed a multi-function rock-gas coupling experimental device and high burst proneness briquette, analyzed the influence effect of gas pressure on burst proneness and failure mode of the specimen, and conclusions drawn from the experiment are reliable. The energy accumulation and dissipation under the experimental condition were also discussed, which provides a certain basis for the occurrence and mechanism of CDDs in coal mines.
However, the following points should be improved before publication:
1、Reference should be further reinforced, especially using of new literature published in a high-quality journal.
2、The new multi-phase (solid-gas coupling) testing facility should be described in detail, such as the technical specifications.
3、In the experiment, the gas adsorption duration is set as 24 hours. For briquette, is it reasonable? How to ensure gas saturation within 24 hours? The authors should provide a discussion on that.
4、The FIG. 9 and 10 should be further improved, and the label of the ordinate should be inward.
5、The author can try to explain the reasons for the relationship between gas pressure and the mechanical strength of coal, which can help readers solve some confusion to some extent.
6、The conclusion should be further condensed, and some results should be supported with experimental data.
Author Response
Attachment

Reviewer 3 Report
In this paper, the failure characteristics and dynamic disaster tendency of coal like briquettes with high explosive tendency under different atmospheric pressures are studied. In the introduction, information is given regarding the necessity of the research carried out, with references to the existing level of knowledge on the studied topic. The research methodology is appropriate and the results are presented in an adequate manner. The writing is standardized, the thinking is clear. But at the same time, I also found some problems in the paper. I hope the author can revise it well to improve the quality of the paper.
(1)The mould consists of a compressive arm, two semi-circular steel tubes, a bottom plate and three fixing devices. Compression arm, semi-circular steel tubes, bottom plate, and fixing devices need to be marked with specific figure numbers in the manuscript.
(2)The different damage pattern in Figure 4 need to be marked.
(3)The name of the subsection in 3.2 needs to be modified.
(4)The mechanism of coal failure mechanism needs to be further explored in detail.
(5)It is recommended that the conclusions be appropriately modified to highlight the research focus and novelty.
(6)Please check the language typing and grammatical errors throughout the manuscript.
Author Response
Attachment

Reviewer 4 Report
Please refer to the attached file. Overall, the author(s) required to provide simple and understanding sentences.

Author Response
Attachment

Reviewer 5 Report
In the article under consideration, Topical and important issues related to the analysis of the effect of gas pressure on the mechanism of destruction of a coal-like briquette are discussed.
The natural system "coal-methane" and the change in its state are difficult to study, and the assessment of its gas-kinetic characteristics, including gas content, based on the theory of sorption, can lead to a significant error, especially if the object of study is a highly gas-bearing coal seam.
The methane abundance of mine workings reflects the level of technogenic impact on the coal rock mass during the excavation of a coal seam. In turn, the quality of forecasting the methane abundance of workings depends on the accuracy of determining two characteristics of the coal seam - its natural gas content and gas kinetic properties. This article focuses on the problem of insufficient accuracy in determining the gas content of a coal seam using existing methods of direct measurements, and provides an example of a technical solution that can improve the measurement accuracy.
The use of the direct method for both core sampling and coal sludge involves the determination of lost gas by the pressure increase over time in special samplers. This approach cannot fully reflect the real gas-kinetic properties of coal due to the influence of the increasing pressure of free gas in the closed space of the sampler, and, consequently, the process of methane desorption slows down. It is possible to solve the problem of determining the true gas-kinetic properties by using the direct sampling method when drilling a hole (well) in an isolated (from the atmosphere of a mine working) mode with measuring the methane consumption from the moment of coal destruction to its sealing in special samplers.
The results obtained in the article will be of interest to readers in the field under consideration.
However, there are the following issues that should be clarified:
1. In the introduction, a more complete review of the literature on emergency situations that occur during the operation of coal mines in various regions of the world should have been given. In particular, the following works could be considered:
Balovtsev S. V., Skopintseva O. V., Kolikov K. S. Aerological risk management in preparation for mining of coal mines. Sustainable Development of Mountain Territories. 2022;14(1):107-116. DOI: 10.21177/1998-4502-2022-14-1-107-116.
Bosikov, I.I.; Martyushev, N. V.; Klyuev, R.V.; Savchenko, I.A.; Kukartsev, V.V.; Kukartsev, V.A.; Tynchenko, Y.A. Modeling and Complex Analysis of the Topology Parameters of Ventilation Networks When Ensuring Fire Safety While Developing Coal and Gas Deposits. Fire 2023, 6, 95. doi: 10.3390/fire6030095.
2. The article should have given a more complete description of the schematic diagram of the test facility, shown in Figure 3. Conduct a comparative analysis of the possibilities of its use in various conditions in different regions of the world.
3. It is not entirely clear from the article whether a wider variation in parameters (diameter, temperature, etc.) is possible for the application of the developed methodology for the sample preparation process (Figure 2).
4. Based on the results shown in Figures 5, 9, 10, mathematical modeling should be carried out with specific regression dependencies that could be used to calculate and predict output values.
5. It should be explained why the absence of gas and gas pressure are taken as an example (Figure 6). How justified is this?
6. The article should have focused on the prospects for using the results obtained in various conditions in various regions of the world.
7. It is not clear from the article whether a patent is supposed to be obtained according to the developed methodology? In my opinion, the findings could be implemented in a patent for a research methodology.
Author Response
Attachment

Round 2
Reviewer 4 Report
All the comments have been addressed by the author(s)
Author Response
Attachment

Reviewer 5 Report
Your answers to my remarks 3 and 7 would be nice to have in the text of the article.
Author Response
Attachment
